# Characterization of *SlBAG* Genes from *Solanum lycopersicum* and Its Function in Response to Dark-Induced Leaf Senescence

**DOI:** 10.3390/plants10050947

**Published:** 2021-05-10

**Authors:** Mingming He, Yu Wang, Mohammad Shah Jahan, Weikang Liu, Abdul Raziq, Jin Sun, Sheng Shu, Shirong Guo

**Affiliations:** 1College of Horticulture, Nanjing Agricultural University, Nanjing 210095, China; 2017204043@njau.edu.cn (M.H.); ywang@naju.edu.cn (Y.W.); shahjahansau@gmail.com (M.S.J.); 18832015631@163.com (W.L.); raziqagrian@gmail.com (A.R.); sunj72@163.com (J.S.); shusheng@njau.edu.cn (S.S.); 2Suqian Academy of Protected Horticulture, Nanjing Agricultural University, Suqian 223800, China; 3Department of Horticulture, Sher-e-Bangla Agricultural University, Dhaka 1207, Bangladesh

**Keywords:** tomato, Bcl-2-associated athanogene, subcellular localization, leaf senescence, gene expression

## Abstract

The Bcl-2-associated athanogene (BAG) family is a group of evolutionarily conserved cochaperones involved in diverse cellular functions. Here, ten putative *SlBAG* genes were identified in tomato. *SlBAG2* and *SlBAG5b* have the same gene structure and conserved domains, along with highly similar identity to their homologs in *Arabidopsis thaliana*, *Oryza sativa*, and *Triticum aestivum*. The qPCR data showed that *BAG2* and *BAG5b* were highly expressed in stems and flowers. Moreover, both genes were differentially expressed under diverse abiotic stimuli, including cold stress, heat stress, salt treatment, and UV irradiation, and treatments with phytohormones, namely, ABA, SA, MeJA, and ETH. Subcellular localization showed that SlBAG2 and SlBAG5b were located in the cell membrane and nucleus. To elucidate the functions in leaf senescence of BAG2 and BAG5b, the full-length CDSs of *BAG2* and *BAG5b* were cloned, and transgenic tomatoes were developed. Compared with WT plants, those overexpressing *BAG2* and *BAG5b* had significantly increased chlorophyll contents, chlorophyll fluorescence parameters and photosynthetic rates but obviously decreased ROS levels, chlorophyll degradation and leaf senescence related gene expression under dark stress. Conclusively, overexpression *SlBAG2* and *SlBAG5b* could improve the tolerance of tomato leaves to dark stress and delay leaf senescence.

## 1. Introduction

Leaf senescence is a fine regulatory mechanism caused by multiple internal factors, such as cell death, plant hormones, nutrient deficiency, senescence related genes, and environmental factors, such as drought, high temperature, salinity, weak light and darkness, and so on [1]. The change of leaf color is the most obvious trait of leaf senescence, and the internal structure of leaves changes obviously during senescence, which is manifested in decreased chlorophyll content and abnormal chloroplast structure [1]. Therefore, the most significant feature of leaf senescence is the process of chlorophyll degradation. Senescence associated genes (SAG), NAC-like (NAP2), and senescence-inducible chloroplast stay-green protein 1 (SGR1) expression levels are increased at the onset of senescence, while expression levels of photosynthesis related genes and chlorophyll biosynthesis related genes are downregulated [2,3]. Due to the negative effects of chlorophyll catabolism enzymes, especially pheophytin pheophorbide hydrolyase (PPH), pheophobide a oxygenase (PAO), chlorophyll a reductase (HCAR), non-yellow coloring 1 (NYC1), and NYC1-like proteins (NOL) result in leaf senescence showing leaf yellowing [4,5]. Transcriptions of chlorophyll catabolism genes are directly relevant to the severity of environmental factors-induced leaf senescence in plants [6,7]. Another characteristic of leaf senescence is excessive accumulation of reactive oxygen species (ROS) [8,9], ROS homeostasis, and redox states regulated growth or senescence related cell death. Leaf senescence induced by darkness is the most effective method to study senescence [10,11].

Bcl-2-associated athanogene (BAG) proteins are a family of multifunctional proteins that regulate apoptosis and tumorigenesis. Members of this family were initially identified during a programmed cell death (PCD) study in mammals and were predicted to be associated with the antiapoptotic protein Bcl-2 to promote cell survival [12,13]. BAG proteins have been characterized as cochaperones [14]. BAG proteins share a functional domain termed the BAG domain, which is composed of three α-helices that are approximately 30–40 amino acids in length and located at the carboxyl terminus of the protein sequence. In addition to the BAG domain, there are other functional domains at the amino terminus of BAG proteins that are associated with metastasis and the subcellular localization of functional proteins [13]. BAG-like family genes have been identified in plants, such as Arabidopsis, soybean, rice, chickpea, potato, sugar beet, cotton, and aspen [14,15]. In Arabidopsis, a total of seven BAG-like proteins were identified, of which four proteins had not only a conserved BAG domain but also a UBQ (ubiquitin-like domain) at the N-terminus, which was very similar to the structure of the human BAG1 protein [16]. Therefore, these proteins may be the direct homologs of mammalian BAG1 in plants [17]. Additionally, plant BAG proteins have a novel CaM (calmodulin-like motif) domain next to the BAG domain, suggesting that plant BAG proteins may have more complicated functions than human BAG proteins.

The BAG family is an evolutionarily conserved, multifunctional group of proteins involved in diverse cellular functions within a variety of physiological processes [18]. Plant BAG proteins can maintain the unfolded protein response of the endoplasmic reticulum and enhance the adaptability of plant cells to heat and cold stress [19]. Kang et al. [20] identified AtBAG6 as a new kind of calmodulin-binding protein and found that oxidizing radicals can induce AtBAG6 transcription and cause PCD in yeast and plants to overexpress *AtBAG6*. The E3 ubiquitin ligase encoded by EBRI and OsBAG4 forms an interactive molecular module that jointly regulates the immune homeostasis of rice and maintains the balance between disease resistance and rice yield [21]. At the transcriptional level, *HSG1* (a member of the *VvBAG* family in grapes) is induced by heat stress; moreover, heterologous *HSG1* overexpression in Arabidopsis can promote flower organ differentiation and produce heat resistance by activating the expression of the *CONSTANS* gene in the photoperiod pathway [22]. Transgenic rice plants overexpressing AtBAG4 showed higher salt resistance than wild-type rice [23]. A new transcriptional complex formed by OsSUVH7, OsBAG4, and OsMYB106 regulated the expression of *OsHKT1;5* (high-affinity K+ transporter), thereby improving salt tolerance under salinity stress [24]. The signal complex formed by CaM, AtBAG5, and Hsc70 can dynamically regulate the darkness-induced leaf senescence process during plant growth and development [25]. Until now, there has been little knowledge of *BAG* family members in solanaceous vegetables, and their functions have not been reported in detail.

Tomato is a model plant for solanaceous crops. With the completion of its genome sequence, the study of gene function has become more important and resulted in a diverse research field. In this study, we identified and characterized ten *BAG* family members in tomato. In addition, we found that the expression of *BAG2* and *BAG5b* was induced by a variety of abiotic stresses and phytohormones. Importantly, overexpression of *BAG2* and *BAG5b* in tomato increased the tolerance of transgenic plants to dark-induced stress and played a role in leaf senescence.

## 2. Materials and Methods

### 2.1. Gene Identification and Bioinformatics Analysis

To identify putative *SlBAG* genes in tomato, BLAST searches of the Sol Genomics Network (https://solgenomics.net) were performed using the amino acid sequences of *AtBAG* genes retrieved from the TAIR (https://www.arabidopsis.org/index.jsp) website. The corresponding gene sequences and amino acid sequences were checked in the Sol Genomics Network and NCBI Database (https://www.ncbi.nlm.nih.gov). The molecular weights and theoretical isoelectric points of the deduced polypeptides were calculated from ProtParam (https://web.expasy.org/compute_pi). Domain information of SlBAG sequences was inferred by querying the Pfam database (http://pfam.xfam.org). Gene structures and conserved domains of SlBAG were plotted using IBS1.0 software. The phylogenetic trees of a set of BAG amino acid sequences were constructed using MEGA software (version 7.0), and the relationships between BAG proteins among species were analyzed by applying the neighbor-joining algorithm with 1000 bootstrap replications. Clustal X2 and Gene Doc software were used to compare the BAG amino acid sequences from different species.

### 2.2. Tomato Materials and Growth Conditions

*Solanum lycopersicum* cv. “Moneymaker” seeds were sown in the nursery substrate after whitening in the sprouting chamber and were cultured in the artificial climate chamber of the Facility Horticulture Research Center of Nanjing Agricultural University under suitable conditions for tomato growth (14 h/10 h photoperiod with 600 μmol m^−2^ s^−1^ light intensity, 23~25 °C/18~20 °C temperature period, and 65~70% relative humidity). Tomato plants were transplanted into pots at the two-leaf stage and irrigated with nutrient solution every alternate day. At the seven-leaf stage, plants were treated with various abiotic stresses, 4 °C as cold stress, 42 °C as heat stress, 150 mM NaCl for root-irrigation as salt stress, and UV as ultraviolet irradiation stress, and hormonal treatments via leaf sprays of 100 μM salicylic acid (SA), 100 μM methyl jasmonate (MeJA), 100 μM abscisic acid (ABA), and 150 µL/L ethephon (ETH). The samples were collected at 0, 1, 3, 6, 12, 24, and 48 h. Roots, stems, old leaves (first euphylla near cotyledon), tender leaves (leaves of growing point), flowers, and fruits were sampled from tomato plants growing in a normal environment to the flowering and fruiting period. All the samples were frozen immediately in liquid nitrogen and stored at −80 °C until use for RNA extraction.

### 2.3. RNA Extraction and qPCR Analysis

RNA was extracted from plant tissue using the RNA Simple Total RNA Kit (Tiangen, China). Each sample was reverse transcribed into cDNA using the HiScript^®^ II QRT SuperMix for qPCR kit (Vazyme, China) following the manufacturer’s protocol. qPCR amplification was performed using the ChamQ SYBR qPCR Master Mix (Vazyme, China) with tomato *actin* used as an internal reference gene and tomato cDNA as a template under different treatment conditions. qPCR involved an initial denaturation (95 °C for 30 s), followed by 40 cycles of 95 °C for 10 s, 58 °C for 30 s, and 72 °C for 30 s, and a final extension at 72 °C for 3 min. The specific primer pairs used in this experiment are listed in Appendix A. The abundance was estimated based on the mean of three biological replications and was calculated using the 2^−ΔΔCT^ method [26].

### 2.4. Subcellular Localization of SlBAG2 and SlBAG5b

The CDSs of *SlBAG2* and *SlBAG5b* without a stop codon were amplified using the primers listed in Appendix A and inserted into the vector pFGC5941-GFP to generate fusion constructs. The generated constructs were introduced into tobacco epidermal cells via Agrobacterium infiltration into *Nicotiana benthamiana* leaves. After 12 h of dark treatment, the plants were kept under normal growth conditions for 2 d. The GFP activity of the materials was monitored using confocal microscopy (LSM 800, Zeiss). The control samples were transformed with an empty vector.

### 2.5. Genetic Transformation of Tomato

The *BAG2* and *BAG5b* overexpression plasmids carrying HA-tag were introduced into the wild-type tomato “Moneymaker” via Agrobacterium-mediated genetic transformation. Tomato transformation was performed according to the Fillatti [27] method, and two independent lines were chosen for testing. First, appropriate PCR vector primers and amplification primers (Appendix A) were used to identify positive lines at the DNA level. Then, the relative expression level was determined by qPCR using *BAG2*-F/R and *BAG5b*-F/R primers. Finally, the protein abundance of BAG2 and BAG5b was detected using western blotting.

### 2.6. Phenotype Analysis of Dark-Induced Stress Tolerance of Transgenic Materials

For the assessment of leaf senescence in transgenic tomato homozygous lines (F2), seven-leaf stage wild-type (WT), *BAG2* OE-10#, *BAG2* OE-12#, *BAG5b* OE-3#, and *BAG5b* OE-13# plants were placed in a light incubator and grown in the dark for 10 d along with 14 h/10 h period with no light and 23~25 °C/18~20 °C temperature period. The third leaves from the apex were collected and used to measure subsequent physiological indicators. Photographs were taken of different lines at 0 d (before treatment) and 10 d (after treatment). The aboveground parts of the plants were cleaned, dried, and weighed for fresh weight. Samples of different lines were put into envelopes, placed into an oven at 105 °C for 15 min, and then dried at 75 °C to a constant weight; the dry weight of the aboveground parts of the tomato plants was recorded.

### 2.7. Chlorophyll Content Detection

Of the third functional leaf (from the top to bottom), 0.2 g was cut into pieces, 20 mL of a mixed solution of acetone:ethanol:water 4.5:4.5:1 (v:v) was added, and the leaves were kept in the dark. The OD_645_ and OD_663_ values were measured after the leaves turned white. The chlorophyll content was calculated according to the following formula:Chl a (mg/g) = (12.7 × OD_663_ − 2.69 × OD_645_) × V(mL)/(1000 × m (g))
Chl b (mg/g) = (22.9 × OD_645_ − 4.68 × OD_663_) × V(mL)/(1000 × m (g))

### 2.8. Determination of Chlorophyll Fluorescence Parameters and Photosynthetic Rate

The maximum photochemical efficiency of PSII (Fv/Fm) and the effective quantum yield of PSII (Y(II)) were measured by an imaging-PAM chlorophyll fluorescence analyzer (Heinz Walz, Effeltrich, Germany) after dark adaptation for half an hour. The photosynthetic rate (Pn) of plants acclimated to light for 30 min was measured using a portable photosynthesizer LI-6400 (LI-COR Inc., USA). A light intensity of 1000 μmol m^−2^ s^−1^, chamber temperature of 25 °C, relative humidity of 70% and CO_2_ concentration of 400 ± 10 μmol mol^−1^ were maintained.

### 2.9. Determination of Hydrogen Peroxide and Superoxide Anion

H_2_O_2_ was detected according to the method of Alexieva et al. [28]. Particularly, the leaf sample was homogenized in 1.6 mL of 0.1% (*w*/*v*) trichloroacetic acid on ice and centrifuged at 12000 rpm for 20 min at 4 °C. A reaction mixture including 0.5 mL of supernatant, 1 mL of 1 M potassium iodide, and 0.5 mL of 0.1 M potassium phosphate buffer (pH 7.8) was maintained in the dark for 1 h, and the absorbance value at OD_390_ was read. The H_2_O_2_ content was calculated by a standard curve prepared with known H_2_O_2_ concentrations. The method described by Elstner and Heupel [29] was used to determine the generation rate of O_2_^•−^.

### 2.10. Protein Extraction and Western Blot Analysis

For protein extraction, 0.3 g fresh weight samples were ground in a mortar with liquid nitrogen and incubated for half an hour in 600 μL of extraction buffer (50 mM Tris-HCl (pH 8.0), 150 mM NaCl, 1 mM EDTA, 1% (*v*/*v*) NP−40 (Nonidet P−40 substitute), 1% (*w*/*v*) sodium deoxycholate, 0.1% sodium dodecyl sulfate (*w*/*v*) and 1 mM phenylmethanesulfonyl fluoride). The extracted proteins were quantified according to the Bradford kit (FD2003, FudeBio, China). The supernatant was boiled with 2 × SDS buffer for 5 min and then subjected to western blotting. Immunoblot analysis was performed to detect the proteins with HA-Tag (M2003, Abmart) (overexpressed plasmids carrying labels) and Rubisco L (AS03037, Agrisera) (as a control) antibodies; these proteins were then incubated with the corresponding secondary antibody. Finally, the protein bands were examined with a gel imager (ChemiDoc Touch, BioRad, Hercules, CA, USA).

### 2.11. Statistical Analyses

The whole experiment was repeated at least three independent biological replicates to analyze each component. For statistical significance analysis, SPSS 22.0 software and Tukey’s honestly significant difference (HSD) test were employed to determine significant differences among the treatments. Differences were considered significant at *P* < 0.05 for all tests.

## 3. Results

### 3.1. Identification of BAG Family Genes in Tomato

To identify *BAG* genes in tomato, we used the amino acid sequences of known *AtBAG* genes as probes to perform BLAST searches in the Sol Genomics Network and NCBI databases. Ten putative *BAG* family members (Appendix A) were identified after surveying the databases. According to the previous names used in NCBI, the tomato BAG family genes were named *SlBAG1*, *SlBAG2*, *SlBAG3a*, *SlBAG3c*, *SlBAG4a*, *SlBAG4b*, *SlBAG5a*, *SlBAG5b*, *SlBAG6*, and *SlBAG7.* These 10 genes were distributed on different chromosomes. The sizes of the *BAG* genes in tomato varied greatly, and the CDS regions of tomato *BAG* family members ranged from 513~3708 bp, among which the shortest sequence was for *SlBAG5b*, coding for 170 amino acids, and the longest sequence was for *SlBAG6,* coding for 1235 amino acids. The physicochemical analysis showed that the molecular weights and theoretical isoelectric points of *SlBAG* genes were within the ranges of 19.4~52.3 kDa and 5.09~10.26, respectively.

### 3.2. Gene Structure and Conserved Motif Analysis of SlBAG Genes

To better understand the structural features, we analyzed the exon-intron structures of *BAG* genes from tomato using IBS.1.0 software. The results showed that the number of introns varied from 0 to 3 in *SlBAG* genes. As shown in Figure 1A, *SlBAG1*, *SlBAG3a*, *SlBAG3c*, *SlBAG4a*, and *SlBAG4b* contained four exons, *SlBAG7* had three exons, *SlBAG5a* and *SlBAG6* had two exons, whereas *SlBAG2* and *SlBAG5b* contained only one exon without an intron. To obtain more insight into the diversification of SlBAG proteins, conserved domains were predicted using the Pfam database. A total of three conserved motifs were predicted across the SlBAG proteins, and the distribution of each motif is shown in Figure 1B. SlBAG1, SlBAG3a, SlBAG3c, SlBAG4a, and SlBAG4b displayed similar motif compositions, which further support highly conservative evolution. For instance, SlBAG1, SlBAG3a, SlBAG3c, SlBAG4a, and SlBAG4b shared ubiquitin-like domains and BAG domains; SlBAG6, SlBAG5a, SlBAG2, and SlBAG5b shared calmodulin-binding motifs and BAG domains; and SlBAG7 contained only the BAG domain. Interestingly, all BAG domains were at the C-terminus of the sequence. Although the biological functions of most BAG genes in plants are unknown, these specific motifs suggest diverse functions of the BAG family in tomato.

### 3.3. Phylogenetic Relationship of Tomato and Other Species

To further understand the relationship between tomato *BAG* genes and those of other species, genetic relationship diagrams of ten tomato SlBAG members, seven Arabidopsis AtBAG members, six rice OsBAG members, and three wheat TaBAG members are shown in Figure 2. We found that tomato BAG proteins shared high identity with their homologs in *Arabidopsis*, *Oryza sativa*, and *Triticum aestivum*. SlBAG1, SlBAG3a, SlBAG3c, SlBA4a, and SlBAG4b were closely related to AtBAG1, AtBAG2, AtBAG3, AtBAG4, OsBAG1, OsBAG2, OsBAG3, OsBAG4, TaBAG1, TaBAG2, and TaBAG3, which were classified as one group, and SlBAG2, SlBAG5a, SlBAG5b, SlBAG6, SlBAG7, AtBAG5, AtBAG6, AtBAG7, OsBAG5, and OsBAG6 were classified as a second group. This result indicated that although BAG members were highly conserved between different species, there were still differences in evolution, suggesting that different SlBAG proteins may be functionally different. In addition, SlBAG2 and SlBAG5b were closely related, implying that the functions of these two BAG proteins were similar.

### 3.4. Amino Acid Sequence Analysis

It is known from the evolutionary tree that SlBAG2 and SlBAG5b were closely related. Clustal X2 and Gene doc software were used to compare the amino acid sequences of SlBAG2 and SlBAG5b with the BAG protein sequences of 12 other species. The C-terminus of these 14 BAG proteins was highly conserved and contained a BAG domain, as shown by the green line in Figure 3, and the N-terminus contained a conserved calmodulin-binding motif (CaM), as shown by the red box in Figure 3, suggesting that CaM acts as a hub connecting calcium signals to the chaperone system.

### 3.5. Temporal Expression of Tomato BAG2 and BAG5b to Diverse Abiotic Stresses and Hormones

The expression profiles of tomato *BAG2* and *BAG5b* under different abiotic stresses, including low temperature, high temperature, NaCl, and ultraviolet radiation stress, and hormone treatments, including SA, MeJA, ABA and ETH treatment, were analyzed by qPCR. Under low temperature conditions, *BAG2* expression in tomato was rather stable during the first 6 h; it then increased and reached a peak at 24 h, but markedly decreased at 48 h (Figure 4A), suggesting that BAG2 was involved in the response to low temperature stress. Under heat stress, the expression level of *BAG2* increased sharply within 1 h and reached 232-fold that of plants without heat stress, and the mRNA abundance at the rest of the time points was the same as that under normal conditions (Figure 4B). The transcription abundance of *BAG2* under NaCl stress was significantly increased at 1 h, then obviously decreased, finally increased again at 48 h (Figure 4C). The expression of *BAG2* was induced during exposure to UV irradiation and reached a maximum at 6 h and 24 h (Figure 4D). With different hormone treatments, the expression of *BAG2* in tomato was induced and showed a similar trend of increasing, then decreasing and finally increasing, but the time point of reaching the peak was slightly different for each hormone (Figure 4E–H).

As shown in Figure 5, *BAG5b* expression in tomato was also affected by abiotic stresses and hormones. Under low temperature stress, similar to *BAG2*, *BAG5b* reached a peak at 24 h, and the expression level was 144-fold higher than that of the control (Figure 5A). The expression level of *BAG5b* dramatically increased after 1 h of heat stress, and the other time points did not change much. This is the same as the result for BAG2 expression, indicating that BAG2 and BAG5b may have similar functions under heat stress (Figure 5B). The expression of *BAG5b* had the same expression trend when subjected to NaCl and UV irradiation stresses, and the maximum expression was approximately 1.7 times that of the control (Figure 5C,D). In addition, the expression of *BAG5b* in tomato reached a maximum at 48 h with SA, JA, ABA, and ETH spraying, while showing a stable trend over time (Figure 5E–H).

### 3.6. Tissue Expression and Subcellular Localization of BAG2 and BAG5b

By analyzing the expression levels of *BAG2* and *BAG5b* in different tissues, it was found that *BAG2* had the highest expression in stems, followed by flowers, and reduced expression in roots, old leaves, and fruits. However, the expression of *BAG5b* was the highest in flowers and stems, followed by roots, with reduced expression in fruits and leaves (Figure 6A,B).

In this study, the subcellular localization of BAG2 and BAG5b was determined by detecting the fluorescence signal of BAG2-GFP and BAG5b-GFP fusion proteins in tobacco through the Agrobacterium-mediated method. As shown in Figure 6C, the green fluorescence of the fusion protein was on the cell membrane and in the nucleus, indicating that BAG2 and BAG5b were located on the cell membrane and present in the nucleus.

### 3.7. Expression Patterns of BAG2 and BAG5b in Leaves of Wild-Type Plants under Dark Treatment

Dark treatment is the simplest and most effective method to induce leaf senescence. We first analyzed the expression patterns of *BAG2* and *BAG5b* in different leaves position of wild-type (WT) plants and different time points under dark treatment. As shown in Figure 7, we found the expression levels of *BAG2* and *BAG5b* in the apex leaves were the highest, and the transcript levels of *BAG2* and *BAG5b* in old leaves were lowest among different positions of leaves. In addition, the levels of *BAG2* and *BAG5b* transcripts were inhibited by dark stress. These results showed that *BAG2* and *BAG5b* may be associated with leaf senescence.

### 3.8. Effect of Dark-Induced Stress on the Growth of Plants Overexpressing BAG

To investigate the role of these two proteins in response to dark-induced leaf senescence, we constructed transgenic tomatoes overexpressing *BAG2* and *BAG5b*. By genomic DNA identification, qPCR and immunoblotting, we screened two independent positive overexpression lines each from the *BAG2* OE and *BAG5b* OE lines (Figure 8) as the samples for subsequent experiments.

As presented in Figure 9A, after 10 d of dark stress, the seedling leaves of *BAG2* OE and *BAG5b* OE plants began to become yellow, which was prominent in WT plant. The fresh and dry weights of the plants overexpressing BAG genes were significantly higher than those of the WT plants suffering from dark-induced stress (Figure 9B,C). As expected, senescence-related physiological attributes, chlorophyll pigment contents were significantly reduced in overexpression plants and WT plants following dark treatment. Chlorophyll a, chlorophyll b and total chlorophyll contents were decreased in WT and *BAG* OE plants when subjected to dark-induced stress. Among, compared with the WT plants, *BAG2* OE and *BAG5b* OE plants displayed higher chlorophyll a and total chlorophyll contents under dark stress. The decrease range of chlorophyll b content in *BAG2* OE and *BAG5b* OE plants after dark treatment was even higher than that in WT plants, so the chlorophyll b content of *BAG2* OE and *BAG5b* OE plants was equivalent to that of WT plants (Figure 9D–F).

### 3.9. Plants Overexpressing BAG Showed Enhanced Chlorophyll Fluorescence Parameters and Photosynthetic Rate

To further analyze the effects of BAG2 and BAG5b under dark-induced leaf senescence, the values of Fv/Fm, Y(II), and Pn of WT and BAG-overexpressing plants were compared. Before dark treatment, Fv/Fm was not significantly different between the overexpression lines and WT. Moreover, Y(II) and Pn of the *BAG2* and *BAG5b* overexpression lines were significantly higher than those in WT plants; after dark treatment, the Fv/Fm, Y(II), and Pn in the leaves of *BAG2* OE and *BAG5b* OE plants were notably higher than those of WT plants (Figure 10). This result indicated that the tolerance of *BAG2* OE and *BAG5b* OE lines to dark-induced stress was significantly higher than that of WT plants.

### 3.10. Effect of Dark-Induced Stress on ROS in WT and BAG-Overexpressing Plants

To investigate the mechanism of BAG in response to dark-induced leaf senescence, the H_2_O_2_ contents and O_2_^•−^ production rates were determined in WT and transgenic seedlings under darkness for 10 d. As shown in Figure 11A, under normal conditions, *BAG2* OE and *BAG5b* OE lines exhibited significantly higher H_2_O_2_ contents than WT plants. After the dark treatment, the H_2_O_2_ content in the leaves of the *BAG2* OE and *BAG5b* OE lines was not obviously different from those in the pretreatment phase, but these values increased sharply in WT plants and were significantly higher than that of the overexpression lines. Before the dark treatment, the O_2_^•−^ production rate in the leaves of the *BAG2* OE and *BAG5b* OE lines was significantly lower than that of the WT plants. The O_2_^•−^ production rate of each line, including the WT, was significantly reduced after dark-induced stress; among them, the O_2_^•−^ level in the overexpressed lines was significantly lower than that of the WT (Figure 11B). These results indicated that *BAG2* OE and *BAG5b* OE lines leaves accumulated lower amounts of ROS compared with WT plants along with the progression of stress duration.

### 3.11. Overexpression of BAG Affected Chlorophyll Degradation and Senescence Associated Genes

Leaf yellowing is the most remarkable change of senescence, leading to leaf chlorophyll degradation mediated by chlorophyll catabolic genes. To analyze the role of BAG2 and BAG5b in dark-induced leaf senescence, the transcriptional levels of chlorophyll degradation related genes (*PPH*, *PAO*, *NOL*, and *NYC1*) and senescence associated genes (*SAG113*, *SAG12*, *SGR1,* and *NAP2*) were monitored. As presented in Figure 12, except for *SAG113*, which showed higher expression levels in overexpression lines, there were no significant differences in chlorophyll degradation genes and senescence genes between WT and *BAG*-overexpressing plants before dark-induced stress treatment. After dark treatment for 10 d, the expression levels of the aforementioned genes in WT plants were remarkably higher than those before treatment, and chlorophyll degradation-related gene and senescence gene expression levels in the leaves of *BAG2* OE and *BAG5b* OE lines were significantly lower than in those of WT plants, indicating that BAG2 and BAG5b delayed chlorophyll degradation and leaf senescence by inhibiting the transcriptional level of chlorophyll degradation- and leaf senescence-related genes.

## 4. Discussion

BAG is a highly evolutionarily conserved auxiliary molecular chaperone that performs multiple functions in diverse environments. BAG proteins are recognized by their common C-terminal conserved domain, known as the BAG domain. In addition to the conserved BAG domain, several other domains have been identified in BAG proteins. These special domains may be related to the regulation of target protein specificity and the localization of BAG proteins in the cell [30]. To date, the majority of reports on *BAG* genes have mainly focused on identification and expression profiles in model plants. However, there is a shortage of literature about the mechanism by which *BAG* genes regulate plant development and function. In the present study, we characterized *SlBAG2* and *SlBAG5b* in tomato in response to different stresses and hormones and analyzed their function under dark-induced leaf senescence.

In previous studies, Zeiner et al. identified six members of the BAG protein family in the human body [31]. Doukhanina et al. identified seven members of the *BAG* family in *A. thaliana* and six *BAG* members in rice [15]. In the current study, we identified and analyzed ten tomato *S1BAG* molecular chaperones on the basis of a database using bioinformatics methods and divided them into two groups according to the phylogenetic relationships shown in the evolutionary tree (Figure 2). All BAG members that have been identified so far are distinguished by a common conserved region located near the C-terminus, termed the BAG domain (Figure 1B), that directly interacts with the ATPase domain of Hsp70/Hsc70, which suggests their high conservation during species evolution [18]. It is found that SlBAG2 and SlBAG5b possessed a CaM domain next to the BAG domain (Figure 1B, Figure 3), indicating that these two genes are evolutionarily identical and similarly functional. Here, this novel CaM domain can regulate the activity of its binding partners by sensing changes in calcium levels and act as the main center between BAG protein and Hsc70 [18]. The phylogenetic analysis portrayed a genetic relationship between SlBAG2 and SlBAG5b and other three species, showing that both BAG proteins were in the same group with AtBAG5, AtBAG6, OsBAG6, OsBAG6, and AtBAG7 (Figure 2). AtBAG5 plays a regulatory role in leaf senescence through Hsc70 mediated signaling pathway [18]. Therefore, we speculate that SlBAG2 and SlBAG5b may also play a role in leaf senescence. In addition, based on gene structure analysis, we found that both *SlBAG2* and *SlBAG5b* had only one exon and no intron, which a different trait from other *BAG* members (Figure 1A), implying BAG proteins might possess redundant but somewhat different functions.

It was noticed that the expression level of *BAG2* in each organ was far higher than that of *BAG5b*, which implicated that the BAG2 played a dominant role in tomato plants. Interestingly, both *BAG* genes were highly expressed in flowers and stems (Figure 6). These results show that BAG2 and BAG5b both perform an important role in plant development, and will guide our future work to figure out whether they play a critical role in flower and stem development. Subcellular localization results showed that SlBAG2 and SlBAG5b were localized to the cytomembrane and nucleus (Figure 5). This localization was not consistent with the results predicted by the website (http://gpcr.biocomp.unibo.it/bacello/pred.htm); however, these results were consistent with the results described by Williams et al. [19] that AtBAG7 was located in the nucleus under heat stress.

We further quantified the expression levels of *SlBAG2* and *SlBAG5b* in response to stress, which laid a foundation for the study of BAG function. Similar to their mammalian counterparts, plant BAG members regulate PCD processes induced by pathogen attack and abiotic stress, speculating that plant BAG responds to the above stress response. Multiple cis-elements related to hormones and stresses have been found in the promoter sequences of *AtBAG4* and *AtBAG6* [18]. In this study, we observed the different expression profiles of *SlBAG2* and *SlBAG5b* in leaves under abiotic stresses, including cold stress, heat stress, NaCl stress, and UV light stress (Figure 4 and Figure 5). Similar to previous studies, the *BAG* gene transcript levels were induced by adversity stress. The overexpression of *AtBAG4* in tobacco exhibited different levels of mRNA abundance under responses to environmental stresses such as UV light, cold, oxidants, darkness and salinity [15]. Wang et al. [24] proposed that the amount of *OsBAG4* in roots was increased in response to salt stress. The peak values of *SlBAG2* and *SlBAG5b* expression were reached at 1 h under heat stress (Figure 4B and Figure 5B) and at 24 h under cold stress (Figure 4A and Figure 5A). These expression patterns may be related to the presence of GC-rich (cold response action element) and HSE (heat shock response component) cis-acting elements involved in cold and heat responses in the promoter sequence of BAG [18]. The effect of NaCl and UV on *SlBAG2* and *SlBAG5b* expression in tomato seedlings appeared to be quite different: the expression of *SlBAG2* had two peaks under salt and UV stress, but the time of occurrence of those peaks was different; however, the expression profile of *SlBAG5b* showed a U-shaped trend under salt and UV stress, indicating that the functions of SlBAG2 and SlBAG5b were different under salt stress and UV treatment. Expression patterns of *SlBAG2* and *SlBAG5b* had been changed by spraying exogenous hormones, including SA, MeJA, ABA, and ETH (Figure 4 and Figure 5). The process of plant PCD is strictly mediated by hormonal signals, such as ethylene, ABA, SA and JA [32,33]. In addition, several recent studies have indicated that effects of plant hormones are symbiotic or opposite throughout leaf senescence [34,35,36]. The expression trends of *SlBAG2* and *SlBAG5b* after treatment with these four hormones were respectively similar. implying that they might be involved in abiotic stress responses in an SA-, MeJA-, ABA- and ETH-dependent manner. However, the specific mechanism in this study still needs further verification. Generally, the plant hormones like SA and ABA interacted in opposites [37], ABA and ETH made analogous effect at different stages of plant growth, development and under different stress conditions [36,38]. Taken together, *SlBAG2* and *SlBAG5b* both responded to hormones and abiotic stresses, suggesting that they may function independently or synergistically during the stress response (Figure 4E–H and Figure 5E–H).

Leaf senescence can reduce crop yield and quality. Therefore, it is of great significance to study the genes functions and mechanisms mediating plant senescence to improve crop performance under natural and stress conditions. A large number of genes are activated during leaf senescence, which is consistent with the importance of transcriptional regulation in leaf senescence and nutrient remobilization [39,40]. Although BAG proteins are involved in a variety of physiological processes, there is little information about their involvement in regulation of leaf senescence to date. Here, we show that BAG2 and BAG5b, with BAG domain protein, which interact with Bcl−2 proteins, and the two synergistically act to improve cell survival, thus improving plant tolerance to stress [18]. In this article, an experiment on the tolerance of transgenic tomato to dark-induced leaf senescence was carried out. Dark-induced senescence first occurred at the top of the plant and resulted in leaf yellowing. Expression of *BAG2* and *BAG5b* was higher in apex (Figure 7B) and the up-regulated expression of *BAG2* and *BAG5b* increased the tolerance to dark induction (Figure 9), which suggests that BAG2 and BAG5b regulates developmental and dark-induced leaf senescence. Dark treatment could downregulate the expression of *BAG2* and *BAG5b* (Figure 7C) and lead to early leaf senescence. Moreover, BAG2 and BAG5b are not limited to senescence, BAG2 and BAG5b may also be involved in regulating plant growth and development rather than organ senescence. In vivo studies showed that *BAG2* OE and *BAG5b* OE plants were larger than WT plants, together with a slight increase in chlorophyll content, Fv/Fm, Y(II), and Pn when grown under dark-induced stress, indicating that the BAG2 and BAG5b proteins confer tolerance to stress (Figure 9 and Figure 10). This is consistent with what has been reported regarding the improvement of stress tolerance in plants overexpressing *BAG.* For example, Pan et al. [41] reported that the activation of *AtBAG6* and *AtBAG7* expression can increase tolerance to salt stress; Ghag et al. [42] reported that *MusaBAG1* expression was strongly induced by *Fusarium oxycomycetes* (Foc) and that transgenic banana plants overexpressing *MusaBAG1* showed superior resistance to banana wilt caused by Foc; Doukhanina et al. [15] found that phenotypes of *AtBAG4* knockout mutants accelerated senescence, which was related to PCD-associated senescence.

Leaf senescence is accompanied by a series of events, such as a decrease in chlorophyll content, chloroplast degradation, and the photosynthetic rate; changes in reactive oxygen species and free radical metabolism; degradation of proteins, nucleic acids, and other biological macromolecules; and a significant change in endogenous plant hormone contents in leaves [43]. Under continuous dark-induced stress, the contents of chlorophyll pigments degraded, the carbohydrate and protein levels in cells decreased, plant senescence occurred, and ROS accumulated in the cells. These processes are precisely controlled by multiple genes and proteins. In Arabidopsis, ROS are considered signaling molecules during leaf senescence [44,45]. Dark-induced senescence is often accompanied by the accumulation of H_2_O_2_ [46]. To further understand the role of BAG in leaf senescence, we examined H_2_O_2_ levels in dark-treated leaves of WT and *BAG*-overexpressing plants. As a result, we found that the basal level of H_2_O_2_ in *BAG*-overexpressing plants was higher than that in WT plants, which was consistent with the results reported by Li et al. [25], presumably caused by decreased free Hsc70 molecules due to the tight association of AtBAG5 with Hsc70 or may indicate that H_2_O_2_ as a ROS signal transduction molecule can mediate the expression of related genes to regulate the growth of tomato. However, following the dark treatment, leaves from BAG transgenic lines exhibited a lower accumulation of H_2_O_2_ than those from WT plants (Figure 11A). These results are agreement with Choudhary et al. [47], suggesting that BAG2 and BAG5b participate in leaf senescence by regulating the production of H_2_O_2_. Moreover, the senescence of leaves was accompanied by the production of O_2_^•−^ [48]. In the present investigation, the production rates of O_2_^•−^ in *BAG2* and *BAG5b* overexpressed plants before dark treatment were lower than that in WT plants, indicating that *BAG2* and *BAG5b* overexpressed plants grew in normal condition negatively regulate the production rate of O_2_^•−^, which may be related to the role of BAG as an anti-apoptotic factor [12]. However, the production of O_2_^•−^ decreased after dark-induced stress (Figure 11B), which was consistent with Mcrae and Thompson [49], who reported that O_2_^•−^ radicals were significantly downregulated in late senescence.

The upregulated expression of *SlBAG2* and *SlBAG5b* occurred with the reduction of ROS in dark-induced condition, which further delayed the process of leaf senescence. In addition, leaf senescence is often accompanied by the upregulation of the expression of certain genes. These genes, which are defined as senescence-associated genes, precisely control the progress of leaf senescence. At the same time, dark-induced stress can also cause chlorophyll degradation, accompanied by changes in chlorophyll-degradation gene expression. Previous studies have confirmed that *SAG113*, *SAG12,* and *NAP2* are upregulated during leaf senescence [50,51], and *SGR1*, *PPH*, *PAO*, *NOL*, and *NYC1* are upregulated act as chlorophyll-degradation inducing genes [52,53,54]. In our study, during darkness, the expression levels of chloroplast degradation-related genes and leaf senescence-related genes in BAG-overexpressing plants were also significantly lower than those in WT plants (Figure 12). According to all the present study results, we speculated that SlBAG2 and SlBAG5b participate in chloroplast degradation and leaf senescence not only through the regulation of ROS production but also through the modulation of related gene expression. Finally, we can confirm that sufficient overexpression of *SlBAG2* and *SlBAG5b* is beneficial for dark-induced stress tolerance. Although many efforts have been made to investigate senescence [39,55], the exact mechanisms underlying this event are not well understood [1]. Our evidence provides a clue for the function of these genes in the regulation of dark-induced, and probably natural, senescence [12,25]. However, the specific regulatory mechanism remains to be verified.

## 5. Conclusions

In summary, this paper provided details on the investigation of tomato BAG proteins. Here, we show that SlBAG2 and SlBAG5b are membrane- and nucleus-localized chaperones that are induced by multiple adverse conditions. Specifically, SlBAG2 and SlBAG5b are integral in the response to dark-induced leaf senescence by downregulating ROS level and leaf senescence-related genes. Taken together, these findings further focus on the importance of the BAG family in plant stress and function in delaying leaf senescence.

## Figures and Tables

**Figure 1 plants-10-00947-f001:**
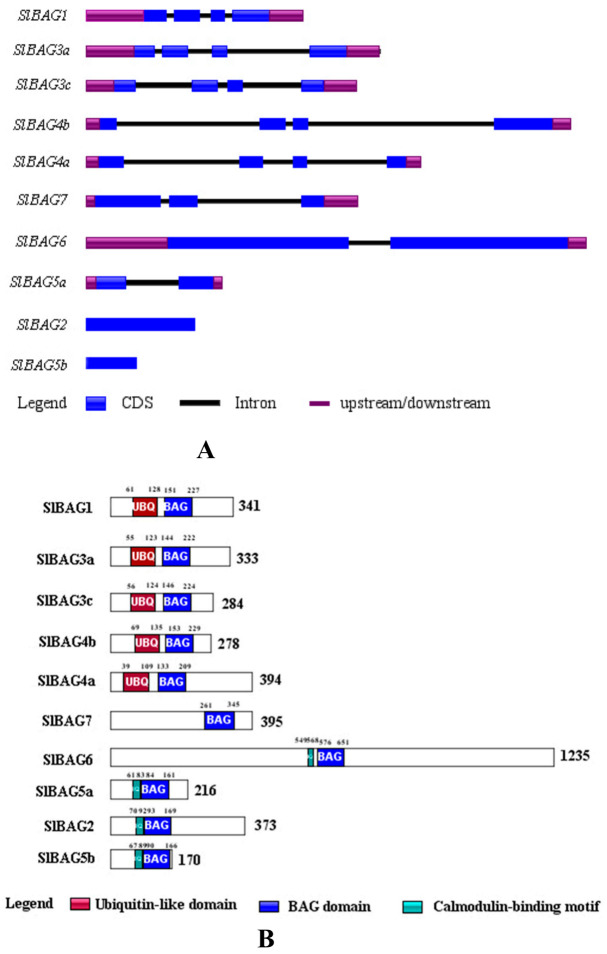
Gene structure and conserved domains of the tomato *BAG* genes. (**A**): Gene structure map. The black lines represent introns. The blue rectangles represent the CDS region. The purple rectangles represent the upstream/downstream regions; (**B**): conserved domain map. Numbers indicate the number of amino acids. UBQ represents the ubiquitin-like domain. IQ represents the Calmodulin-binding motif.

**Figure 2 plants-10-00947-f002:**
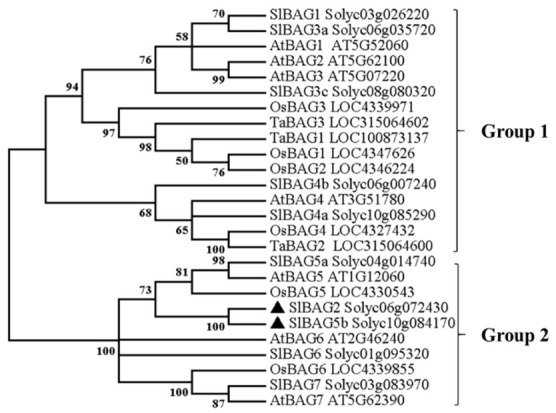
Phylogenetic tree of tomato BAG and BAG proteins from other species. *Solanum lycopersicum* (Sl), *Arabidopsis thaliana* (At), *Oryza sativa* (Os), *Triticum aestivum* (Ta). Based on the phylogenetic tree, these genes were divided into two groups. The black triangles represented SlBAG2 and SlBAG5b.

**Figure 3 plants-10-00947-f003:**
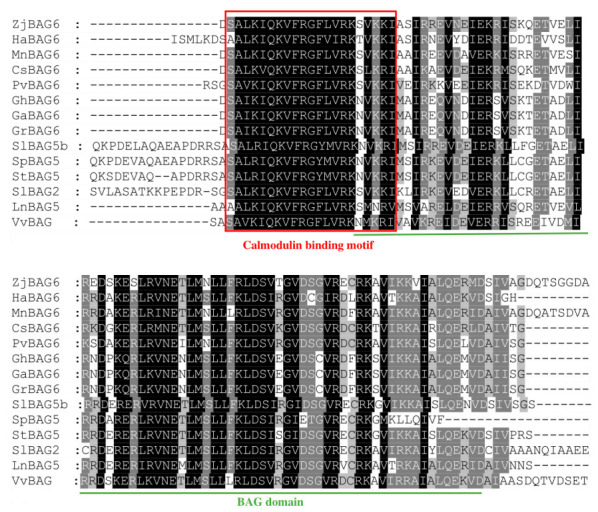
Alignment of SlBAG2 and SlBAG5b amino acid sequences with those of 12 other species: *Solanum tuberosum* (StBAG5; LOC107061352), *Solanum pennellii* (SpBAG5; LOC107001644), *Ipomoea nil* (LnBAG5; LOC109168976), *Morus notabilis* (MnBAG6; LOC21400474), *Ziziphus jujuba* (ZjBAG6; LOC107417637), *Pistacia vera* (PvBAG6; LOC116131206), *Cannabis sativa* (CsBAG6; LOC115714697), *Gossypium raimondii* (GrBAG6; LOC105794789), *Vitis vinifera* (VvBAG; LOC100260747), *Gossypium hirsutum* (GhBAG6; LOC107886709), *Gossypium arboretum* (GaBAG6; LOC108455800), and *Helianthus annuus* (HaBAG6; LOC110904115).

**Figure 4 plants-10-00947-f004:**
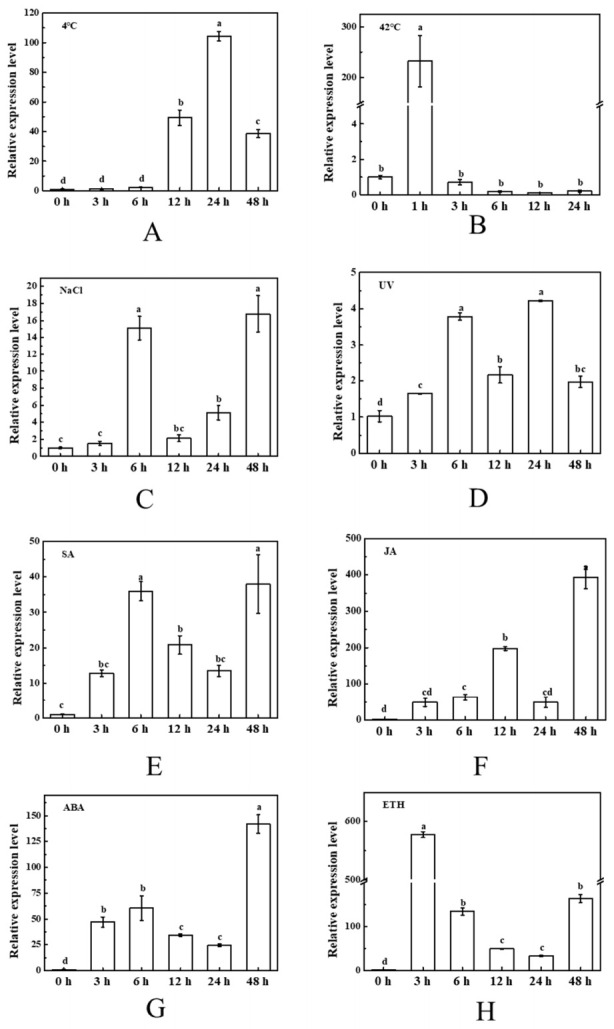
Responses of tomato *BAG2* to different abiotic stresses and various hormones. Abiotic stresses: 4 °C (**A**), 42 °C (**B**), NaCl treatment (**C**), and UV irradiation (**D**). Hormone treatments: SA foliage spraying (**E**), JA foliage spraying (**F**), ABA foliage spraying (**G**), and ETH foliage spraying (**H**). Seedling growth conditions set as 14 h/10 h photoperiod with 150 μmol m^−2^ s^−1^ light intensity, 23~25 °C/18~20 °C temperature period (except 4 or 42 °C). The young leaf samples were collected at the indicated time points and analyzed by qPCR. For each treatment, the expression level at 0 h was set as 1.0. Different letters indicate expression levels that were significantly different at *P* < 0.05.

**Figure 5 plants-10-00947-f005:**
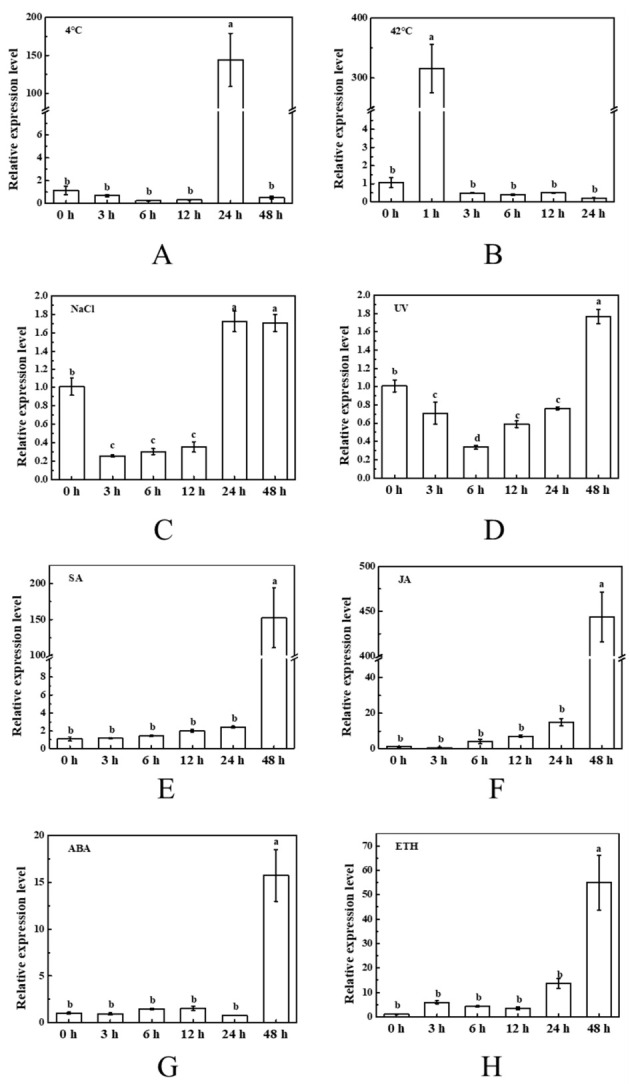
Responses of tomato *BAG5b* to different abiotic stresses and various hormones. Abiotic stresses: 4 °C (**A**), 42 °C (**B**), NaCl treatment (**C**), and UV irradiation (**D**). Hormone treatments: SA foliage spraying (**E**), JA foliage spraying (**F**), ABA foliage spraying (**G**), and ETH foliage spraying (**H**). Seedling growth conditions set as 14 h/10 h photoperiod with 150 μmol m^−2^ s^−1^ light intensity, 23~25 °C/18~20 °C temperature period (except 4 or 42 °C). The young leaf samples were collected at the indicated time points and analyzed by qPCR. For each treatment, the expression level at 0 h was set as 1.0. Different letters indicate expression levels that were significantly different at *P* < 0.05.

**Figure 6 plants-10-00947-f006:**
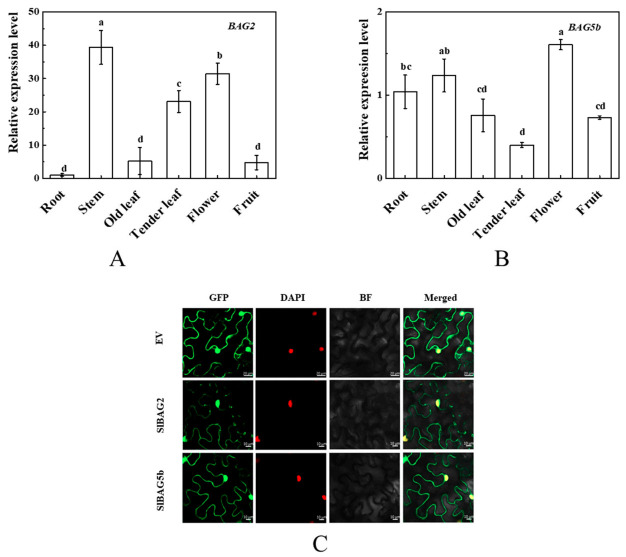
Tissue-specific expression analysis of *BAG2* and *BAG5b* in tomato and subcellular localization. (**A**,**B**): The expression levels of *BAG2* and *BAG5b* in the root, stem, old leaf, tender leaf, flower, and fruit of tomato. The expression level in the root was set as 1.0. Data represent the means ± SE of three replicates. Different letters represent expression levels that were significantly different at *P* < 0.05; (**C**): subcellular localization of BAG2 and BAG5b in tobacco cells. The CDS of *BAG2* and *BAG5b* without a stop codon was inserted into the vector pFGC5941-GFP. The empty vector (EV) was used as a control.

**Figure 7 plants-10-00947-f007:**
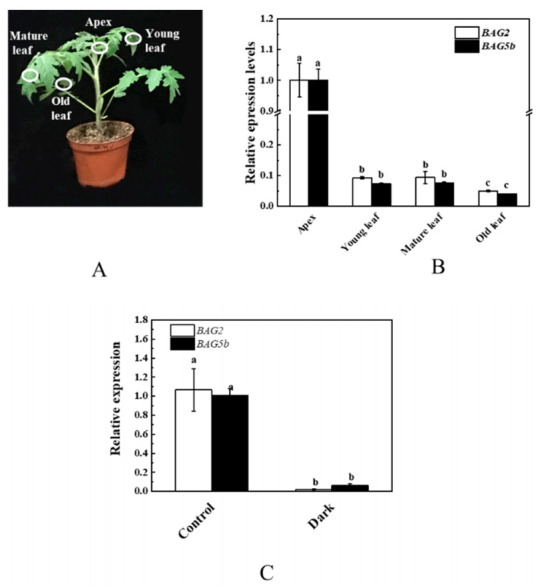
Expression pattern analysis of *BAG2* and *BAG5b* in wild-type (WT) plants under dark treatment. (**A**): Different parts of seedlings before dark stress; (**B**): levels of *BAG2* and *BAG5b* transcripts were determined by qPCR in apex, young leaf, mature leaf, old leaf of WT plants developed in dark stress for 10 d. For both genes, the expression level in the apex was set as 1.0; (**C**): *BAG2* and *BAG5b* expression in leaves treated with darkness was detected by qPCR. The transcripts of both genes in 0 d were set as 1.0. Different letters indicate expression levels that were significantly different at *P* < 0.05.

**Figure 8 plants-10-00947-f008:**
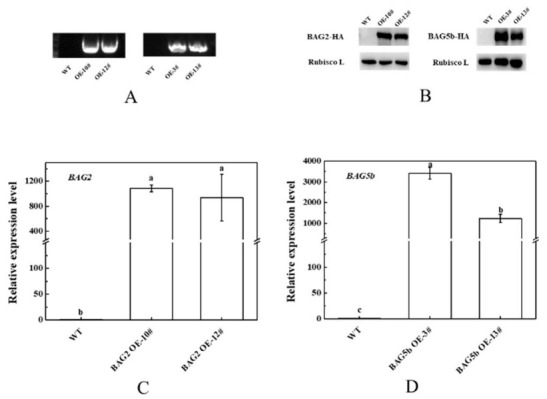
Overexpression of *BAG2* and *BAG5b* in transgenic tomato. (**A**): Genome PCR; (**B**): Western blot; (**C**,**D**): qPCR test. OE−10# and OE−12# were two independent samples of plants overexpressing *BAG2*. OE−3# and OE−13# were two independent samples of plants overexpressing *BAG5b*. For each gene, the expression level of the WT plant was set as 1.0. Different letters indicate expression levels that were significantly different at *P* < 0.05.

**Figure 9 plants-10-00947-f009:**
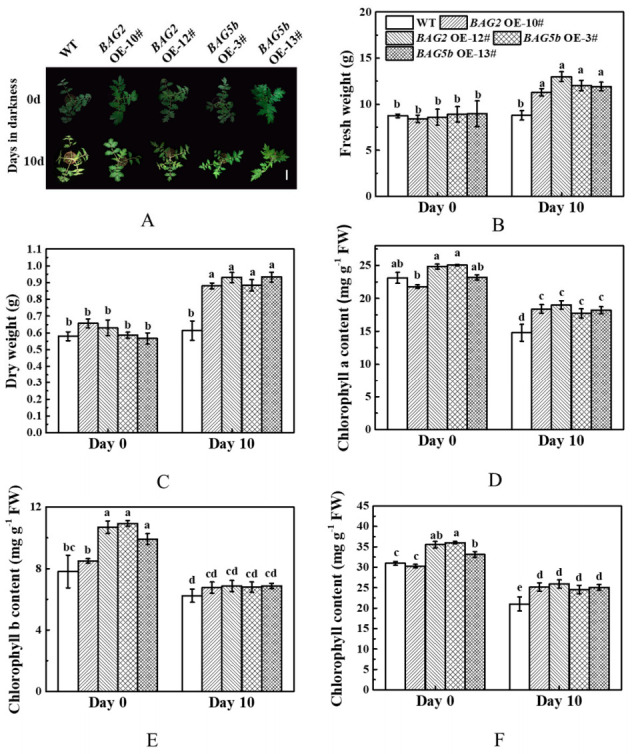
Effects of dark stress on the growth of WT and plants overexpressing *BAG2* or *BAG5b*. (**A**): Photographs of plant phenotypes were taken 0 d and 10 d after dark stress; (**B**): fresh weight; (**C**): dry weight; (**D**–**F**): chlorophyll content. Different letters indicate that each indicator was significantly different at *P* < 0.05. 10# and 12# represent 2 lines of *BAG2* OE plants; 3# and 13# represent 2 lines of *BAG5b* OE plants; OE, overexpression; WT, wild type. Bars = 10 cm.

**Figure 10 plants-10-00947-f010:**
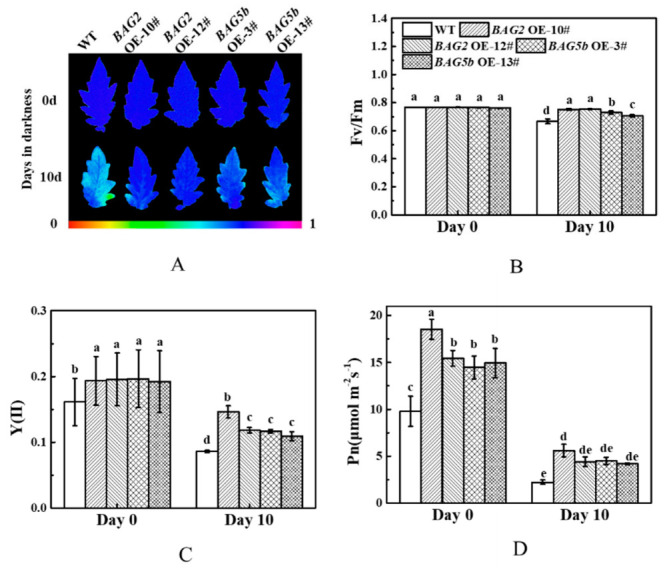
Effects of dark stress on fluorescence and photosynthesis of WT, *BAG2*, and *BAG5b* overexpressing plants. (**A**): Pseudocolor images showing the maximum quantum efficiency of PSII (Fv/Fm); the color code in the images ranges from 0 (black) to 1.0 (purple). (**B**): The values of Fv/Fm in the terminal leaflets of the fourth leaves before treatment and after 10 d of darkness. (**C**,**D**) The values of Y(II) and Pn in the terminal leaflets of the fourth leaves before treatment and after 10 d of darkness. Different letters indicate significant differences at *P* < 0.05. 10# and 12# represent 2 lines of *BAG2* OE plants; 3# and 13# represent 2 lines of *BAG5b* OE plants; OE, overexpression; WT, wild type.

**Figure 11 plants-10-00947-f011:**
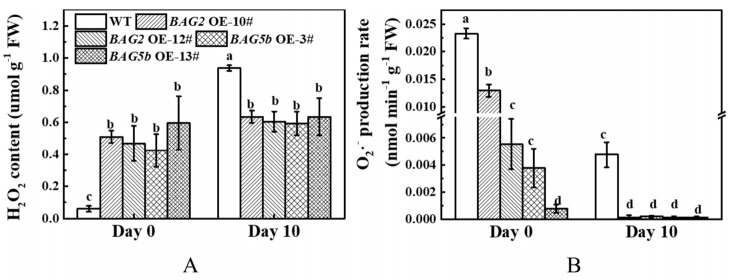
Effects of dark stress on ROS in WT, *BAG2* OE, and *BAG5b* OE plants. (**A**): The H_2_O_2_ content was detected in the fourth leaves after 0 d and 10 d of darkness. (**B**): The O_2_^•−^ production rate was detected in the fourth leaves after 0 d and 10 d of darkness. Different letters indicate significant differences at *P* < 0.05. 10# and 12# represent 2 lines of BAG2 OE plants; 3# and 13# represent 2 lines of BAG5b OE plants; OE, overexpression; WT, wild type.

**Figure 12 plants-10-00947-f012:**
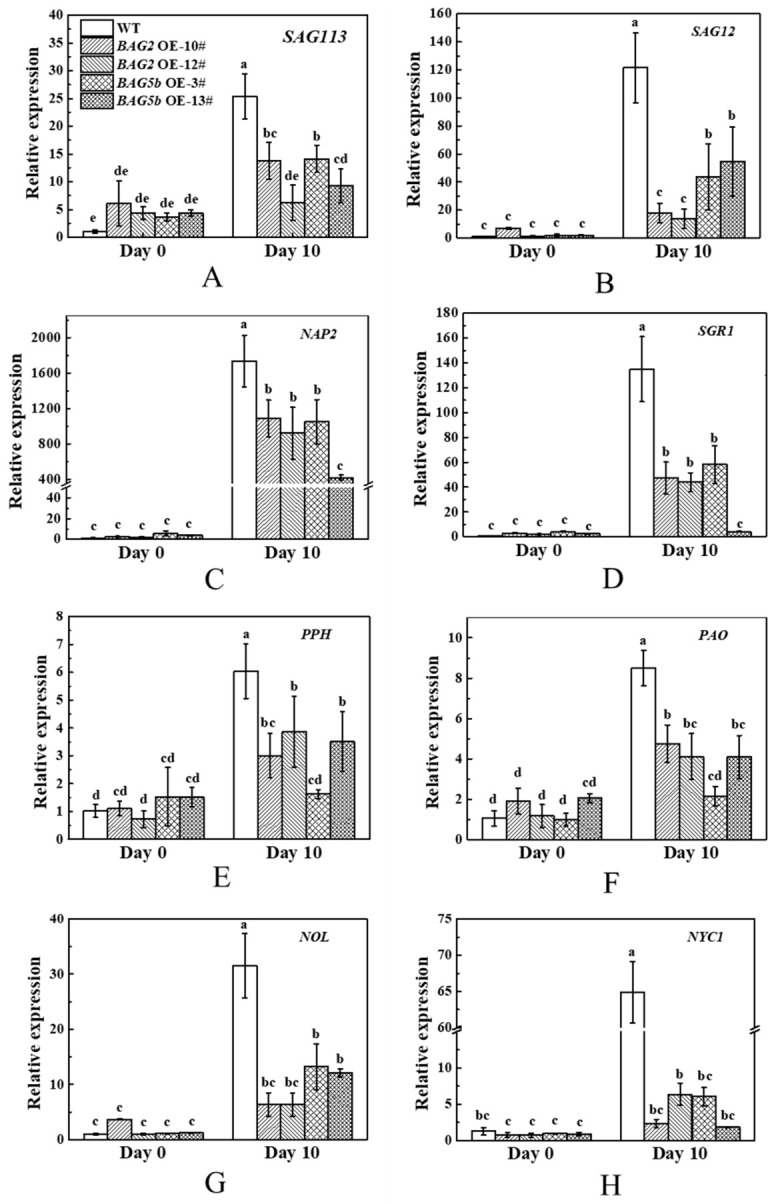
Expression of senescence related genes ((**A**): *SAG113*, (**B**): *SAG12*, (**C**): *NAP2*, and (**D**): *SGR1*) and chlorophyll degradation genes ((**E**): *PPH*, (**F**): *PAO*, (**G**): *NOL,* and (**H**): *NYC1*) in WT, *BAG2* OE, and *BAG5b* OE transgenic lines under dark stress. The young leaf samples were collected at the indicated time points and analyzed by qPCR. For each gene, the expression level of WT plants at 0 d was set as 1.0. Different letters indicate that the expression levels were significantly different at *P* < 0.05. 10# and 12# represent 2 lines of *BAG2* OE plants; 3# and 13# represent 2 lines of *BAG5b* OE plants; OE, overexpression; WT, wild type.

## Data Availability

Data sharing not applicable.

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
