# Peer review of "Characterization of SlBAG Genes from Solanum lycopersicum and Its Function in Response to Dark-Induced Leaf Senescence"

_plants, 2021, doi:10.3390/plants10050947_

Round 1

Reviewer 1 Report

The manuscript identified ten BAG genes in tomato, and among them, the authors have studied two of them: SlBAG2 and SlBAG5b. The manuscript demonstrate that expression of both genes was upregulated in response to abiotic stress and hormone treatments. The authors produced overexpressing transgenic tomatoes lines, which showed delayed leaf senescence. Overall, the manuscript is clear, the methodology appropriated and the obtained results interesting. However, I have some points that could be improved.

  1. Lettering in Figure 1, regarding the domains is very small.
  2. The meaning of triangles in Figure 2 is not indicated in Figure legend.
  3. It is not clear the reason to choose SlBAG2 and SlBAG5b to further analysis, there are two other genes with the same domains in tomato, and one of them (SlBAG5a) is also in the same branch.
  4. In response to hormones, a different pattern for both genes can be deduced, SlBAG2 is induced in response to hormones with apparent biphasic responses, whereas SlBAG5b is induced in the latter time points. Any suggestion on that?
  5. Something is wrong with legend to Figure 7.
  6. The title for Figure 10 is confusing, I found only photosynthetic parameters whereas it is stated: “Effects of dark stress on the growth of…”.
  7. I disagree with the sentence “Expression of BAG2 and BAG5b was higher in yellowing 411 leaves (Figure7B) and increased in response to dark-induced stress (Figure 9)” (lines 411-412). This is not indicated in those figures.
  8. A clear difference is obtained regarding relative expression of both genes in different tissues, SlBAG2 showed huge difference whereas SlBAG5b seem to be constitutive (Figure 6). This is not discussed.
  9. I agree with the final sentence in discussion, but I consider that some more recent references could be used

Reviewer 2 Report

The manuscript “Characterization of SlBAG genes from tomato and its function in response to dark induced leaf senescenceprovide the details on the investigation of tomato BAG proteinsspecifically, SlBAG2 and SlBAG5b are integral in the response to dark-induced leaf senescence by related genes. These findings showed the importance of the BAG family in plant stress and function in delaying leaf senescence.

To sum up, this article can find an interest for the specialists in this field after taking account of the following minor corrections.

Title: The title is good and reflecting the study properly. use the scientific name of tomato in the title.

Abstract:

The abstract is fine and well demonstrated the whole study.  

Keywords: The keywords are appropriate.

Introduction

The introduction is fine and according to the study. However,correct the following minor mistakes.

Line 333-35: Add reference of the statement “The change of leaf color is the most obvious trait of leaf senescence, and the internal structure of leaves changes obviously during senescence, which is manifested by decreased chlorophyll content and abnormal chloroplast structure.”

Line 39: Add comma between “--------- photosynthesis related genes” and “chlorophyll biosynthesis related genes-----" 

Materials and Methods

Methodology is well designed and accurate.

Results and Discussion

The results and discussion are fine and justified with scientific reasoning.

Conclusions

The conclusion is well summarized.

References

References are not properly formatted. Kindly follow the Instructions for Authors.

Reviewer 3 Report

The manuscript deals with BAG genes and proteins in combination with an effect of abiotic stress factors, exogenous phytohormones and dark-induced senescence of tomato plants including transgenic plants overexpressing BAG2 and BAG5b. The topic is interesting and actual, the results presented are new. However, the manuscript suffers from certain shortcomings. Some of methods and plant samples and treatments are not described sufficiently (see comments below). The results are not discussed comprehensively, the interconnection of the findings obtained in the individual experiments is relatively weak and therefore there is no more comprehensive view of the function of the studied BAGs in tomato plants. The Discussion should therefore be rewritten.

Comments:

  1. It is not clear what leaves were analysed by qPCR in experiments with abiotic stress factors and phytohormones’ treatment (Fig. 4 and Fig. 5). It is important as expression level of BAGs was different in leaves of different age (Fig. 7). Could the authors state how they explain the different changes in BAG2 and BAG5b expression levels? There is no specification of the conditions under which the treatments took place (e.g. light conditions, temperature - if it was not 4 or 42°C, etc.). Similarly, conditions during dark treatment are not described.
  2. It is not stated how long did the dark treatment of wt plants take when the expression pattern analysis was performed (Fig. 7). The Fig. 7C does not correspond to the information in the figure’s legend. On page 15, the authors described a different leaf yellowing induced by dark treatment, but this cannot be deduced from the Fig. 7.
  3. It is not true, that the chlorophyll b content “remained relatively stable” in BAG2 OE and BAG5b OE plants (Fig. 9E), the decrease in chlorophyll b content after dark treatment is even higher than in wt plants.
  4. The formulation “enhanced chlorophyll fluorescence parameters” is not suitable as in case of some parameters, their increase indicates a worse function of photosystem II (or photosynthetic function or fitness generally). I recommend using basic interpretation of the presented chlorophyll fluorescence parameters (maximal efficiency of PSII in case of Fv/Fm, etc.). The definition of Y (II) parameter is missing as well as used measuring parameters of PAM analyser.
  5. I recommend using another scale of the “color code” in Fig. 10A (Fv/Fm imaging) – for example, from 0.5 to 0.85 – it should highlight differences among the leaves.
  6. In Fig. 12, again it is not clear what leaves were used for analysis.
  7. Line 495: the irradiance should not be given in lx, but in µmol of photons m-2 s-1.
  8. A duration of acclimation of the plants on leaf chamber conditions before measuring the photosynthetic rate (Pn) should be stated (line 555 – 558).

Round 2

Reviewer 1 Report

The authors have addressed correctly the points raised in the previous version of the manuscript.